Factors constraining natural recovery of Diadema antillarum following a mass die-off: a case study near the island of Saba, Caribbean Netherlands

Hylkema Alwin alwin.hylkema@hvhl.nl
Klokman Oliver J.
1 Van Hall Larenstein University of Applied Sciences , Leeuwarden , Netherlands
2 Marine Animal Ecology Group, Wageningen University and Research , Wageningen , Netherlands
Vidjak Olja
Electronic publication date: 2025 Dec 17
Publication date: 2025
Volume: 13
Electronic Location ID: e20418
Received 2025 Mar 12; Accepted 2025 Oct 28
Copyright: ©2025 Hylkema and Klokman
Copyright year: 2025
Copyright holder: Hylkema and Klokman
License: This is an open access article distributed under the terms of the Creative Commons Attribution License, which permits unrestricted use, distribution, reproduction and adaptation in any medium and for any purpose provided that it is properly attributed. For attribution, the original author(s), title, publication source (PeerJ) and either DOI or URL of the article must be cited.
License URL: https://creativecommons.org/licenses/by/4.0/

Keywords: Sea urchin, Echinoid, Settlement, Predation, Shelter availability

Funding: RAAK-PRO Diadema I and Diadema II RAAK.PRO03.005 RAAK.PRO05.025 SIA, part of the Netherlands Organization for Scientific Research (NWO) This study was conducted within the scope of the RAAK-PRO Diadema I and Diadema II projects (project# RAAK.PRO03.005 and RAAK.PRO05.025), funded by SIA, part of the Netherlands Organization for Scientific Research (NWO). The funders had no role in study design, data collection and analysis, decision to publish, or preparation of the manuscript.

==============================
After the mass mortality of 1983–1984, recovery of the keystone herbivore Diadema antillarum has been limited. Persistently low population densities reduced grazing pressure, leading to algal dominance on many Caribbean reefs. To understand recovery dynamics and support restoration, greater insight into factors constraining sea urchin population recovery is essential. We assessed D. antillarum settlement, predator density, shelter availability, and post-die-off recovery at four locations near Saba, Caribbean Netherlands, following the 2022 D. antillarum die-off. One location, an artificial rock reef, had previously supported high D. antillarum densities, while the natural reefs showed only sporadic recruitment. One-year post-die-off, the D. antillarum density at the artificial reef recovered to 1.4 ± 0.5 D. antillarum per m2, whereas no populations established on the three natural reefs. Despite high overall predator biomass, the artificial reef had lower densities of Spanish hogfish and small wrasses. Previous studies indicate that these species, together with the queen triggerfish, are key determinants of D. antillarum recovery. Invertebrate predator densities were broadly similar across locations, although spotted spiny lobster, an important crab predator and potentially reducing crab predation on D. antillarum, were more abundant on the artificial reef. Shelter availability, depth, and reef structure appeared more favorable at the artificial reef location. We conclude that D. antillarum population establishment is primarily shaped by predation pressure and shelter availability, while larval settlement and the presence of adults appear less influential. A regional-scale study could further detangle the factors affecting natural recovery and identify reefs with a high chance of successful D. antillarum restoration.

Introduction

The long-spined sea urchin Diadema antillarum (Philippi 1845) suffered two Caribbean wide mass mortalities in 1983–1984 (Lessios, 2016) and in 2022 (Hylkema et al., 2023). Before the 1983–1984 die-off, which reduced population densities by 98% (Lessios, 2016) D. antillarum was the most abundant herbivore on Caribbean coral reefs (Lessios, Garrido & Kessing, 2001). Herbivorous fishes, the only other abundant group of herbivores, were already severely overfished in most of the Caribbean (Pandolfi et al., 2003) and were not able to compensate for the loss of grazing pressure after the D. antillarum mass mortality (Levitan, 1988; Carpenter, 1990a). In the years following the D. antillarum mass mortality, turf and macroalgae became the dominant benthic groups (DeRuyter van Steveninck & Bak, 1986; Hughes, Reed & Boyle, 1987; Carpenter, 1990b).

Additional threats, particularly climate-induced heatwaves and diseases, caused widespread coral mortality in subsequent decades (Riegl et al., 2009), further opening space for algal expansion. Turf and macroalgae compete with corals for light, space, and nutrients (McCook, Jompa & Diaz-Pulido, 2001). The proliferating algae inhibited coral recruitment and reduced the resilience of Caribbean coral reefs (Mumby, Hastings & Edwards, 2007), which resulted in a stepwise degradation which continues today.

The cause of the 1983–1984 mass mortality event in D. antillarum remains unidentified (Lessios, 2016). In contrast, the scuticociliate Philaster apodigitiformis, was identified as the causative agent of the 2022 die-off (Hewson et al., 2023). In the year following this event, similar mortalities were reported in other diadematid sea urchins across the Mediterranean, Red Sea, and Western Indian Ocean (Zirler et al., 2023; Ritchie et al., 2024; Roth et al., 2024). The spatial and temporal patterns of these outbreaks suggest that the scuticociliate spread via natural and anthropogenic vectors, such as maritime traffic, and was likely absent from these regions prior to the die-offs (Ritchie et al., 2024). However, the origins of the scuticociliate and the environmental or ecological factors triggering its proliferation remain poorly understood and it remains to be determined how the sea urchin populations recover.

After the 1980s die-off, D. antillarum recovery has been very poor, with populations estimated at around 12% of their former densities in 2015 (Lessios, 2016). This slow recovery, combined with the 2022 die-off, has prolonged the functional extinction of D. antillarum on most Caribbean reefs. The slow D. antillarum recovery is likely the result of a multitude of factors. Initially, Allee effects might have prevented effective fertilization (Lessios, 1988). In addition, the removal of many populations resulted in downstream reductions in D. antillarum larval densities and settlement rates (Bak, 1985). More recent studies focusing on D. antillarum settlement on artificial substrates demonstrated a lack of larval influx continued to constrain D. antillarum recovery in certain locations (Miller et al., 2009; Feehan et al., 2016). However, settlement rates in other locations were in the same order of magnitude as pre-die-off densities on Curaçao (Williams, García-Sais & Capella, 2009; Vermeij et al., 2010; Hylkema et al., 2022a; Klokman & Hylkema, 2024). At these locations, high post-settlement and post-recruitment mortality, driven by predation (Harborne et al., 2009; Hylkema et al., 2022b) and limited shelter (Bodmer et al., 2021), likely constrained D. antillarum recovery.

Understanding the extent to which settlement rates, predation pressure, and shelter availability affect D. antillarum recovery is essential to select suitable restoration locations. However, all these factors correlate with the abundance of adult D. antillarum. Adults create a suitable habitat for juvenile conspecifics, for example by creating shelter for juveniles with their spine canopy (Miller et al., 2007). The intense grazing of adults creates bare substrate covered with a fresh biofilm, which is both an important settlement cue (Wijers et al., 2024a) and an important food source for the settlers (Vermeij et al., 2010). Conversely, reefs without adult D. antillarum provide less suitable habitat for settlers. Without the intense grazing of adults, turf and macroalgae fill small shelter spaces and reduce shelter availability (Spadaro & Butler, 2021) while at the same time offering shelter to D. antillarum micro-predators (Bechtel, Gayle & Kaufman, 2006) that reduce post-settlement survival.

The 2022 D. antillarum die-off, with a lethality of 99% on the locations included in this study (Hylkema et al., 2023), provided the opportunity to study factors affecting natural recovery with minimal correlating effects of adult populations. The aim of this case study was to investigate the effect of settlement rates, predation pressure, and shelter availability on the recovery of D. antillarum after the 2022 die-off at four locations near Saba, Caribbean Netherlands.

Methods

Locations

All research was conducted at four dive sites: Diadema City, Tent Reef, Ladder Bay, and Torrens Point, located at the southwestern side of Saba, Caribbean Netherlands (Fig. 1). These locations were chosen because they have been monitored for D. antillarum settlement since 2019 (Klokman & Hylkema, 2024). The dive sites were located within the Saba Marine Park and permission for this study was given by the Saba Conservation Foundation, management authority of the Saba Marine Park. Diadema City (Fig. 2) consists of a former breakwater which was destroyed and turned into an artificial reef by hurricane Hugo in 1989. The breakwater was made of locally available rock in a variety of sizes. Diadema City is 100 m long and 10–20 m wide. Three semi-permanent 30 m transects were established in a row from east to west on hard substrate at 8.1, 7.9 and 7.0 m depth. Tent Reef consists of a reef plateau running east to west along the shore at 4–8 m deep. Three semi-permanent 30 m transects were established on the reef plateau at 5.8, 6.1 and 5.6 m depth. Ladder Bay mostly consists of encrusted boulders of volcanic origin. Three semi-permanent transects were established over these boulders, aiming for hard substrate, at 10.0, 8.4 and 7.4 m depth. Torrens Point consists of several lava fingers of 5–10 m wide and up till 50 m long, interspersed with volcanic boulders. Semi-permanent transects were established on the lava fingers at 9.4, 8.9 and 7.6 m depth.

Figure 1 Map of study locations.

Map indicating the four locations near Saba, Caribbean Netherlands, and the location of Saba in the Caribbean region. Map created with ArcMap 10.8 using data from Esri and Garmin.

D. antillarum populations

Prior to the 2022 die-off, Diadema City contained the largest D. antillarum population around Saba (Hylkema et al., 2023). At Tent Reef, Ladder Bay, and Torrens Point, D. antillarum were occasionally observed prior to the 2022 die-off, but large populations were absent. D. antillarum populations were assessed at each location approximately one month after the 2022 D. antillarum die-off, in the period March–May 2022, and again one year after the die-off in March 2023. At each location, all D. antillarum within the three 30 × 2 m semi-permanent transects were counted by two scuba-diving researchers.

Figure 2 Four research locations.

Overview photos of the four research locations in 2023: Diadema City (A), Tent Reef (B), Ladder Bay (C), and Torrens Point (D). Photos were taken at 6–8 m depth.

Settlement rates

D. antillarum settlement rates were determined monthly during the D. antillarum settlement season, which runs from April–October in this part of the Caribbean (Hylkema et al., 2022a; Klokman & Hylkema, 2024), in 2022. At each location a sub-surface buoy and anchor were used to vertically keep a rope with pre-made loops at 8.0, 8.5, 9.0, 9.5 and 10.0 m in the water column, following Klokman & Hylkema (2024). A settlement collector, consisting of 15 bio-balls strung on a nylon fishing line, was attached to each loop. Every month, the five settlement collectors deployed at each location were replaced with new collectors. Collected bio-balls were analyzed in the lab for D. antillarum settlers by carefully rinsing them in plastic trays following Hylkema et al. (2022a).

Fish predator density and presence

To determine the density of fish predators, three monitoring dives were conducted per location in the period May–October 2023. D. antillarum predators were selected from previous studies using fish stomach content analyses (Randall, Schroeder & Starck, 1964; Randall, 1967) and included all fishes with D. antillarum or other echinoid material in their stomach (Table 1). At every survey, fish predators were counted and size estimated on all three semi-permanent 30 × 2 m belt transects per location. The researcher was swimming with a constant speed of around 6 min per transect using scuba.

After completion of the belt transects, a 20 min roving diver survey was conducted by the same researcher to determine fish predators in a wider area (around 100 × 50 m). The roving diver survey gave the opportunity to include shy species which typically are not observed in belt transects, such as queen triggerfish, and to include additional habitat such as ledges and sandy areas. During the roving diver survey, a researcher covered the indicated area in a zig-zag pattern swimming at a constant speed. All fishes from Table 1 were recorded and size estimated, except for bluehead and yellowhead wrasse, two very abundant small wrasses which were well represented within the belt transects.

Table 1 List of fish species that are known to predate on Diadema, based on Randall, Schroeder & Starck (1964); Randall (1967), and the percentage echinoid material in their stomach based on Randall (1967).

Common name	Scientific name	Echinoid material in stomach (%)	
Queen triggerfish	Balistes vetula	72.8	
Ocean triggerfish	Canthidermis maculata	25.0	
Spanish hogfish	Bodianus rufus	14.4	
Puddingwife	Halichoeres radiatus	19.9	
Slippery dick	Halichoeres bivittatus	17.9	
Yellowhead wrasse	Halichoeres garnoti	3.0	
Black-ear wrasse	Halichoeres poeyi	6.8	
Bluehead	Thalassoma bifasciatum	1.5	
Porcupinefish	Diodon hystrix	34.6	
Spotted trunkfish	Lactophrys bicaudalis	10.0	
Smooth trunkfish	Lactophrys triqueter	2.3	
Black margate	Anisotremus surinamensis	53.5	
Spanish grunt	Haemulon macrostomum	86.8	
Caesar grunt	Haemulon carbonarium	10.9	
French grunt	Haemulon flavolineatum	1.5	
Bluestriped grunt	Haemulon sciurus	8.7	
White grunt	Haemulon plumierii	12.4	
Jolthead porgy	Calamus bajonado	45.2	
Saucereye porgy	Calamus calamus	8.9	
Bandtail puffer	Sphoeroides spengleri	6.9	
Sharpnose puffer	Canthigaster rostrata	3.8	

Invertebrate predator presence and density

Invertebrate predators were surveyed during three nocturnal monitoring dives per location in the period May–October 2023. These dives were conducted at least 1 h after sunset using scuba. During each survey, macro and micro invertebrate predators were counted and size estimated on the three 30 × 2 m semi-permanent transects. Macro-invertebrate predators of D. antillarum include the spiny lobster Panulirus argus (Randall, Schroeder & Starck, 1964), the spotted spiny lobster P. guttatus (Kintzing & Butler, 2014), the king helmet Cassis tuberosa (Levitan & Genovese, 1989), the queen helmet Cassis madagascariensis (Randall, Schroeder & Starck, 1964) and the batwing crab Carpilius corallinus (Sharp & Reckenbeil, 2022). Little is known about micro-predators of D. antillarum, but it can be assumed that most smaller crustaceans and fireworms predate on sea urchin settlers (Scheibling & Robinson, 2008; Jennings & Hunt, 2011; Cano et al., 2024) so these groups were included in the survey. During the monitoring dives, a single researcher inspected all shelter spaces along the transect using a Bigblue AL1300WP video light, recording all invertebrate predators.

Shelter availability

Shelter availability was determined using a modified version of the Point Intercept Contour (PIC) device modelled after Yanovski, Nelson & Abelson (2017). The PIC device (Fig. 3) reflects deviations from an artificial horizon by inserting 21 one-meter shafts at five cm intervals, allowing detection of shelter spaces at ecologically relevant spatial scales for D. antillarum (Bodmer et al., 2021). The PIC device was deployed every 3 m on each of the 30 m semi-permanent transects, resulting in nine deployments per transect and 27 deployments per location. At every deployment, the PIC device was placed on the reef, all shafts were inserted, and a photo was made from the side.

Figure 3 Point intercept countour device.

Deployed point intercept countour device modelled after Yanovski, Nelson & Abelson (2017). Shafts are 1m in length and marked areas are 10 cm. Shelter for Diadema was defined as at least 5 cm deep and 5–15 cm wide (Bodmer et al., 2021). Based on these criteria, two shelters were identified on this photo (white arrows).

Analysis

Generalized Linear Mixed Models (GLMMs) with a Poisson distribution (glmer function with family = Poisson in the R package “lme4” (Bates et al., 2015)) were used to test the effect of fixed factors year and location on the D. antillarum density. Transect_ID was added to the model as a random factor, to account for the fact that the same transects were surveyed in 2022 and 2023. Model selection was done based on Zuur et al. (2009) and Bolker et al. (2009). The model with the lowest Aikaike information criterion (AIC) was the model including both year, location, and their interaction. Model validation revealed that the model was not overdispersed. Wald χ2 tests were performed for statistical inference of the fixed factors (Bolker et al., 2009), using the Anova function of the R package “Car”.

Generalized Linear Models (GLMs) with a Poisson distribution (glm function with family = Poisson in the R package “lme4”) were used to test the effect of fixed factors location, month and depth on the D. antillarum settlement rate per month. Model selection was done based on Zuur et al. (2009). The model with the lowest AIC was the model including only location and month, without depth or any interaction. Model validation revealed that the model was not overdispersed. Statistical inference was performed with likelihood ratio tests using the Anova function.

Fish predation pressure per survey and transect was calculated using abundance data per species (density per 100 m2) together with known length–weight relationships (Bohnsack & Harper, 1988) to estimate predator biomass per species. Biomass values were then adjusted by the average fraction of echinoid material in the stomach contents of each species, as reported by Randall (1967). This represents a slight adaptation of the method by Harborne et al. (2009), who instead weighted biomass estimates by the proportion of individuals with D. antillarum in their stomachs. Although Randall’s study was conducted at a time when D. antillarum were far more abundant, the relative proportion of echinoid material provides a useful indication of prey preference among species. Species with historically high D. antillarum consumption are still expected to exert greater predation pressure compared to species with low consumption, even if absolute prey availability has changed. This inference is supported by recent studies of predation on restocked D. antillarum, which identified the same species as in the 1960s as key D. antillarum predators (De Breuyn et al., 2023; Wijers et al., 2024b).

The predation pressure per species per 100 m2 was summed to get a total predation pressure per transect. Linear Mixed Models (LMMs, lmer function in the R package “lme4”) were used to test the effect of fixed factors location, survey number and their interaction on the fish predation pressure calculated from the transect surveys. Survey had a value of 1–3, as three surveys were conducted on all three transects of a location. To account for this dependency, transect_ID was added to the model as a random factor. Initial models showed a strong mean to variance relationship, which was solved by cube-root transforming the data. Model selection revealed that the model with location as only fixed factor had the lowest AIC (Zuur et al., 2009). For statistical inference, an F-test with Kenward-Roger’s approximation to degrees of freedom was performed.

For the invertebrate predators, a distinction was made between micro-predators (<3 cm) and macro-predators (>3 cm). Counts per taxonomic group were summed to get total micro- and macro-invertebrate densities per transect. GLMMs with a Poisson distribution were used to test the effect of fixed factors location and survey number on the micro- and macro-invertebrate abundance. Survey had a value of 1–3, as three surveys were conducted on all three transects of a location. To account for this dependency, transect_ID was added to the model as a random factor. Model selection was done based on Zuur et al. (2009) and Bolker et al. (2009). The model with the lowest AIC was the model including both location, survey number and their interaction for micro-predators and only location for macro-predators. Model validation (Zuur et al., 2009; Bolker et al., 2009) revealed substantial overdispersion for the micro-predator model, which was solved using a negative-binomial error distribution (glmer.nb fuction in the R package “lme4”). Statistical inference was performed using likelihood ratio tests conducted with the drop1 function for the micro-predator model and with Wald χ2 tests conducted with the Anova function for the macro-predator model (Bolker et al., 2009).

Photos from the PIC deployments were analyzed using ImageJ software. Based on Bodmer et al. (2021), we counted shelters that were at least 5 cm deep and 5–15 cm wide. Thus, negative deviations of the shafts to neighboring shafts greater than five cm depth and consisting of one to a maximum of three shafts were considered a shelter (Fig. 3). A one-way ANOVA, followed by Tukey post-hoc tests, was performed to compare shelter availability and depth per location. Model validation revealed non-normal distributed residuals and heterogeneity of variances for shelter depth, which was solved with a cube root transformation.

For LMM, GLMs and GLMMs, pairwise comparisons with a Tukey adjustment for multiple comparisons were conducted to examine significance of location using estimated marginal means (EMM) from the package “emmeans”. To compare the D. antillarum density between years per location, pairwise comparisons were conducted with location grouped within year. All analyses were done with R version 4.3.0 using Rstudio 2023.3.1.446. Graphs were made with the package “ggplot2”. Provided values are means ± standard error, while P-values <0.05 were considered statistically significant.

Results

D. antillarum populations

Location (χ2 = 23.6, df = 3, P < 0.001), year (χ2 = 7.5, df = 1, P = 0.006) and their interaction (χ2 = 23.4, df = 3, P < 0.001) had a significant effect on the D. antillarum population. After the 2022 die-off, D. antillarum densities were less than 0.03 ± 0.02 D. antillarum per m2 at all four locations and did not differ among each other (Fig. 4A). One year later, in 2023, D. antillarum densities were very similar, except for Diadema City, where D. antillarum densities had increased to 1.4 ± 0.5 D. antillarum per m2. This was significantly higher than all other locations (P < 0.001 for all comparisons), which did not differ significantly.

D. antillarum settlement

Over the course of the study, 175 D. antillarum settlers were collected from the bio-ball collectors. Location (LRT = 77.7, df = 3, P < 0.001) and month (LRT = 217.9, df = 5, P < 0.001) had a significant effect on the number of D. antillarum settlers per collector (Fig. 4B). Average settlement at Diadema City was 0.1 ± 0.1 D. antillarum per collector per month, which was significantly lower compared to all other locations (P < 0.001 for all comparisons). Settlement at Ladder Bay was 2.6 ± 0.7 D. antillarum per collector per month, which was significantly higher than the 1.3 ± 0.4 found at Tent Reef (P = 0.003), while settlement at Torrens Point (1.5 ± 0.4 D. antillarum per collector per month) did not differ from either Tent Reef or Ladder Bay.

Figure 4 Results per location.

Per location, the average (±SE) (A) Diadema antillarum density (ind m−2) in 2022 (right after the die-off) and 2023, (B) D. antillarum settlement (ind collector−1), (C) fish predation pressure (gr 100 m−2), (D) fish predation pressure (gr roving diver survey−1), (E) invertebrate micro-predator abundance (ind 100 m−2), (F) invertebrate macro-predator abundance (ind 100 m−2), (G) shelter availability (number m−1), and (H) shelter depth (cm). Locations which do not share a common lowercase letter differ significantly from each other.

Fish predators

Fish predator abundance ranged from 128 fishes per 100 m2 at Diadema City to 490 fishes per 100 m2 at Tent Reef (Table 2). Ladder Bay and Torrens Point had intermediate fish abundances. The small wrasses T. bifasciatum and H. garnoti contributed most to the predator abundance at all locations and explained most of the differences between locations. Predation pressure, in this study calculated as the biomass of D. antillarum predators on the transects multiplied by the fraction of their stomach content consisting of echinoid remains (Randall, 1967), ranged from 25 ± 9 g per 100 m2 at Diadema City to 143 ± 73 g per 100 m2 at Torrens Point (Fig. 4C). Survey (not included in best fitting model) and location (F = 1.4, df = 3, P = 0.260) did not significantly affect the predation pressure per m2. At Diadema City, Ladder Bay and Torrens Point, grunts (Haemulidae), specifically Caesar grunt H. carbonarium, black margate A. surinamensis, and, to a lesser extent, French grunt H. flavolineatum, contributed most to the predation pressure (Table S2). At Tent Reef, wrasses (Labridae), specifically Spanish hogfish B. rufus, puddingwife H. radiatus, and bluehead T. bifasciatum, contributed most to the predation pressure.

The results of the roving diver surveys were used to calculate predation pressure of the wider area, which ranged from 390 ±  175 g per survey at Tent Reef to 2,596 ± 498 g per survey at Diadema City (Fig. 4D). As roving diver predation pressure was the result of three surveys of the same area per location, statistical inference was not possible. At Diadema City, the predation pressure per survey was dominated by the black margate, followed by the Caesar grunt (Table S3). At Tent Reef, the Spanish hogfish contributed most to the predation pressure, while at Ladder Bay and Torrens Point both grunts and Spanish hogfish contributed to the predation pressure. During the roving diver surveys only one species, the porcupinefish D. hystrix, was observed in addition to the species already observed on the transects.

Table 2 Abundance of Diadema fish predators (ind 100 m−2, ± SE) for seven species with the highest predation pressure (Table S2), six other encountered species, and in total per location. Species were sorted based on their overall predation pressure.

Average abundance of fish predators (ind 100 m −2 , ± SE)	
Common name	Scientific name	Diadema City	Tent reef	Ladder Bay	Torrens point	
Caesar grunt	H. carbonarium	0.6 ± 0.3	0.0 ± 0.0	0.9 ± 0.5	3.1 ± 2.2	
Black Margate	A. surinamensis	0.0 ± 0.0	0.0 ± 0.0	0.6 ± 0.4	0.2 ± 0.2	
Spanish hogfish	B. rufus	0.0 ± 0.0	2.2 ± 0.8	0.7 ± 0.4	0.4 ± 0.2	
Puddingwife	H. radiatus	0.2 ± 0.2	0.2 ± 0.2	0.2 ± 0.2	0.2 ± 0.2	
Bluehead	T. bifasciatum	109.8 ± 20.9	449.8 ± 88	176.7 ± 20.7	233.7 ± 21.2	
Yellowhead wrasse	H. garnoti	11.9 ± 4.2	36.9 ± 31.4	11.9 ± 4.9	14.1 ± 7.5	
French grunt	H. flavolineatum	5.0 ± 1.6	0.0 ± 0.0	0.0 ± 0.0	1.5 ± 1.1	
6 other species		0.4 ± 0.4	0.9 ± 0.4	0.4 ± 0.2	0.6 ± 0.3	
Total		127.9 ± 22.1	490 ± 74.8	191.4 ± 20.4	253.8 ± 19.5	

Invertebrate predators

Micro-predator (<3 cm) abundance (Fig. 4E) was significantly affected by location (LRT = 389.0, df = 3, P < 0.001), survey number (LRT = 6.2, df = 2, P = 0.046), and the interaction between location and survey number (LRT = 90.4, df = 6, P < 0.001). Post-hoc testing revealed that micro-predator abundance was highest at Diadema City, where 902 ± 147 micro-predators were recorded per 100 m2. This was significantly higher compared to Tent Reef, Ladder Bay, and Torrens Point (P < 0.0001 for all comparisons). At Tent Reef 59 ± 12 micro-predators per 100 m2 were recorded, which was significantly more than the 31 ± 10 at Ladder Bay (P = 0.003) and the 24 ± 9 at Torrens Point (P < 0.001). Locations Ladder Bay and Torrens Point did not differ among each other. The majority of the micro-predators at each location were shrimp (Table S4) and the large difference between Diadema City and the other locations is explained by the much higher shrimp abundance at Diadema City.

Macro-predators (>3 cm) abundance (Fig. 4F) was not significantly affected by location (χ2 = 7.8, df = 3, P = 0.051) or survey number (not included in best fitting model) and ranged from 0.9 ± 0.3 macro-predators per 100 m2 at both Tent Reef and Ladder Bay to 2.6 ± 0.7 macro-predators per 100 m2 at Diadema City. Spotted spiny lobster P. guttatus contributed most to the macro-predators abundance, followed by hermit crabs, Caribbean spiny lobster P. argus, and the king helmet C. madagascariensis (Table S4).

Shelter availability and depth

Shelter availability differed significantly per location (F = 20.6, df = 3, P < 0.001, Fig. 4G). Average shelter availability was with 1.3 ± 0.2 and 1.2 ± 0.1 shelter per meter highest at Torrens Point and Diadema City respectively. These locations did not differ significantly from each other but had significantly higher shelter availability compared to the 0.3 ± 0.1 shelter per m found at Tent Reef. Ladder Bay, with 0.8 ± 0.1 shelter per m did not differ from any of the other locations.

Shelter depth differed significantly per location (F = 4.0, df = 3, P < 0.01, Fig. 4H) and was highest at Diadema City where average shelter depth was 18.6 ± 2.0 cm. This was significantly higher compared to Tent Reef (P = 0.034) and Torrens Point (P = 0.0384), but not to Ladder Bay. These three locations all had average shelter depths around 10–12 cm.

Discussion

After the 2022 D. antillarum die-off, all study locations had very low D. antillarum densities. One year later, only Diadema City supported a substantial population, while densities at the other sites remained similar to the year before. Whether the population at Diadema City will continue to grow remains uncertain, as historical cases show variable trajectories. Following the 1983–1984 die-off, D. antillarum populations in Panama initially increased but later declined when recruitment remained low (Lessios, 1988). Across the wider Caribbean, recovery since the 1980s has been patchy, with only a few locations showing sustained rebounds (Carpenter & Edmunds, 2006; Debrot & Nagelkerken, 2006), which did not translate into large-scale recovery. The recovery at Diadema City demonstrates that the presence of adults is not always necessary for population establishment. Previous studies found that experimentally (Lessios, 1995) or naturally (Levitan, Edmunds & Levitan, 2014) elevated adult densities did not enhance recruitment, although these outcomes may also reflect limited larval supply. Our results show that recovery is possible without substantial adult populations, supporting the conclusions of Lessios (1995) and Levitan, Edmunds & Levitan (2014) that habitat modifications created by grazing adults do not strongly influence recruitment, because all locations had a similar cover of turf and macroalgae in the months after the 2022 die-off. The rapid recovery at Diadema City, combined with the absence of recovery at the other locations, suggests that other site-specific factors, such as larval supply, predation pressure, and shelter availability, are more decisive for population establishment.

Settlement rates on artificial substrates provide an indication of larval supply. Around Saba, settlement rates in 2022 were in the same order of magnitude as in previous years (Hylkema et al., 2022b; Klokman & Hylkema, 2024), likely reflecting spawning that occurred before populations were impacted by the scuticociliate responsible for the mass mortality (Hewson et al., 2023). At Diadema City, settlement was significantly lower than at the other three locations, indicating that larval influx was not the primary driver of population recovery. This pattern is consistent with earlier studies showing high settlement on reefs lacking adult populations (Williams, Yoshioka & García Sais, 2010; Bodmer et al., 2015; Hylkema et al., 2022a; Klokman & Hylkema, 2024). The 2022 die-off killed 99% of the D. antillarum at Diadema City and surrounding reefs (Hylkema et al., 2023, personal observation, 2023, both authors), ensuring that the gross majority of D. antillarum recorded in 2023 were derived from the relatively low settlement in 2022. To our knowledge, this represents the first documented case of D. antillarum population recovery under such low settlement rates.

Fish predation pressure on semi-permanent transects was lowest at Diadema City, although differences with the other locations were not significant. In contrast, the roving diver survey indicated the highest predation pressure there. This discrepancy reflects differences in species composition. At Diadema City, grunts dominated predation pressure, while wrasses were more prominent at the other reefs. Wrasses are diurnal, disperse across the reef during the day and were frequently observed on the semi-permanent transect. Grunts are nocturnal and shelter in overhangs and caves during the day. Such habitats were absent from the semi-permanent transects but abundant in the boulder structure of Diadema City, which supported large schools of black margate and Caesar grunt therefore increasing the predation pressure estimated by the roving diver survey results.

Predation pressure was estimated by combining species biomass with the fraction of echinoid content in their diet (Randall, 1967). This method has limitations, as it assumes dietary preferences remained unchanged since the 1960s (Harborne et al., 2009). After the 1983–1984 die-off, many fishes shifted to alternative prey (Reinthal, Kensley & Lewis, 1984; Robertson, 1987), and it is unclear if they still target D. antillarum (Lessios, 1988). Recent studies suggest that former main predators such as queen triggerfish and Spanish hogfish still actively prey on D. antillarum, while no predation by grunts was recorded (De Breuyn et al., 2023; Wijers et al., 2024b). The apparent role of grunts in our predation metrics may therefore be inflated. Spanish hogfish were almost absent at Diadema City but common at the other reefs, suggesting their density may be an essential factor constraining recovery.

The calculation method also does not distinguish between D. antillarum size classes. Some predators, such as wrasses, target small juveniles (Lessios, 1988), whereas others, like queen triggerfish, prefer larger individuals (De Breuyn et al., 2023). The loss of many juveniles may remove little biomass but strongly limit population recovery. Thus, the role of wrasses in suppressing D. antillarum may be underestimated (Lessios, 1988; Rodríguez-Barreras et al., 2015). Blueheads and yellowhead wrasses were the most abundant predators at all locations, but their density at Diadema City was two- to fivefold lower, which may have facilitated recovery. This reduced abundance could be linked to higher densities of mid-sized piscivores such as snappers and trumpetfish observed at Diadema City.

More broadly, these patterns illustrate how predator–prey interactions and trophic cascades on degraded reefs can shape the recovery of invertebrate herbivores. Around Saba, fishing is primarily recreational and targets large piscivores such as snappers and groupers. Such selective removal of apex predators may trigger mesopredator release (Ritchie & Johnson, 2009), elevating populations of mid-sized predators such as wrasses and increasing predation pressure on D. antillarum. Conversely, Harborne et al. (2009) demonstrated that overfishing outside marine reserves can indirectly facilitate D. antillarum recovery by reducing the abundance of their predators. Together, these findings suggest that D. antillarum recovery can be induced under contrasting scenarios: either through severe overfishing that reduces predator populations or through intact apex predator assemblages that suppress mesopredators, highlighting the complex, context-dependent role of trophic dynamics in reef restoration.

Invertebrate micro-predators (<3 cm), such as shrimps, crabs, and fireworms, are often assumed to inhibit D. antillarum recovery (Williams, García-Sais & Yoshioka, 2011; Hylkema et al., 2022b; Cano et al., 2024), although the magnitude of their effect remains uncertain (Bechtel, Gayle & Kaufman, 2006). In this study, Diadema City had a significantly higher micro-predator abundance, driven entirely by shrimp densities. Shrimps were the most numerous micro-predators at all locations but were 20–30 times more abundant at Diadema City, largely due to red night shrimps (Cinetorhynchus manningi) sheltering and foraging under adult D. antillarum spines. During 2023 surveys, most adults hosted 5–10 individuals that retracted when disturbed (personal observation, 2023, both authors). Dietary preferences of red night shrimps and other Rhynchocinetidae are poorly known, they are likely opportunistic feeders on detritus and sessile invertebrates (Dumont et al., 2009). Although predation on D. antillarum settlers cannot be entirely excluded, the D. antillarum recovery at Diadema City makes it unlikely that they have a large negative effect.

Other micro-invertebrates such as crabs (Harrold et al., 1991; Scheibling & Robinson, 2008; Clemente et al., 2013) and fireworms (Simonini et al., 2017) are known to be important predators of juvenile other sea urchin species. However, on our transects, few of these micro-predators were encountered. Their apparent scarcity may reflect cryptic behavior, suggesting that transect surveys underestimate true densities. Alternative methods, such as baited or refuge traps, could provide more representative abundance estimates (Osawa et al., 2015; Moreira-Ferreira et al., 2020).

Macro-invertebrate predators (>3 cm) observed included spotted spiny lobster, Caribbean spiny lobster, hermit crabs, and king helmet. Abundance did not differ significantly among locations, though Diadema City supported high numbers of both lobster species in large crevices, sometimes co-occurring with adult D. antillarum. Caribbean spiny lobsters forage mostly off-reef and are unlikely to significantly affect adult urchins (Kintzing & Butler, 2014; Randall, Schroeder & Starck, 1964). Spotted spiny lobsters forage on the reef and can predate smaller D. antillarum (Kintzing & Butler, 2014), yet their high densities at Diadema City did not prevent population recovery. This may reflect overestimation of their predation impact or mitigation via consumption of crabs, their primary prey (Butler & Kintzing, 2015), suggesting that high spotted spiny lobster abundance could paradoxically reduce net predation pressure on D. antillarum.

Shelter availability and depth varied considerably among and within locations. Diadema City and Torrens Point exhibited the highest shelter availability, whereas Diadema City and Ladder Bay had the deepest shelters. These characteristics may have facilitated the recovery of D. antillarum at Diadema City, consistent with previous findings that habitat complexity correlates positively with urchin densities (Bodmer et al., 2015; Bodmer et al., 2021). Shelter availability for D. antillarum is difficult to quantify precisely, and our measures of shelter density and depth represent approximations. Moreover, widespread reef degradation has reduced scleractinian coral cover, overall reef complexity, and consequently, natural shelter availability (Alvarez-Filip et al., 2009; Magel et al., 2019).

A potentially critical, yet unquantified, parameter is shelter shape. At Diadema City, wedge-shaped shelters formed by two or more stacked rocks were abundant, whereas such structures were rarely observed at other sites. These wedge-shaped shelters may functionally replace those previously provided by plate-forming corals such as Agaricia sp. or Orbicella sp. (Bodmer et al., 2015). Unlike rounded cup-shaped shelters, which accommodate only specific size classes, wedge-shaped shelters provide protection for a broad range of sizes. Settlers can retract fully to the wedge tip and gradually emerge as they grow, remaining continuously protected. We hypothesize that the combination of high shelter availability, substantial depth, and favorable shelter shape contributed to the successful D. antillarum recovery at Diadema City.

The rapid recovery of D. antillarum at the artificial reef following the 2022 die-off indicates that population establishment is strongly influenced by specific habitat characteristics and not by the presence of adult conspecifics. Although overall fish predator densities were high, Spanish hogfish and smaller wrasses, key species affecting urchin recovery alongside queen triggerfish, were less abundant than at natural reefs, potentially reducing recruit mortality. Invertebrate predator densities were generally similar across sites, with the exception of spotted spiny lobsters, which were more common at the artificial reef and may have indirectly reduced crab predation on D. antillarum. Shelter availability and depth were more favorable at the artificial reef, and the high abundance of wedge-shaped shelters likely further facilitated urchin survival. Settlement rates were lowest at the artificial reef, indicating that settlement density is only a minor driver of population establishment. Reduced shelter availability, possibly combined with increased predation of wrasses possibly constrained recovery on natural reefs where the species used to be abundant before the 1980s die-off.

A broader, regional approach covering multiple islands and reef types could help disentangle factors driving recovery. Future studies should differentiate predation by wrasses and queen triggerfish, quantify micro-invertebrate predation using refuge traps, and include shelter shape metrics if possible. Identifying constraints on natural recovery is critical for selecting effective restocking sites and enhancing the success of D. antillarum restoration, thereby contributing to Caribbean reef resilience.

Supplemental Information

Supplemental Information 1 Diadema density, abundance of Diadema fish predators, and abundance of Diadema invertebrate predators

Supplemental Information 2 Raw data

We are grateful to the SCF staff, especially Marijn van der Laan, for their assistance in the field and to Rosemarijn van de Lint, Jasper Raijmakers, Bai Klap, Jelle van der Velde, Jethro van’t Hul, Pauline Raimbault, and Peter Johnson who helped with collecting the data. We also acknowledge the feedback of two reviewers, which helped us improving this manuscript.

Additional Information and Declarations

Competing Interests

Author Contributions

Field Study Permissions

Data Availability

The authors declare there are no competing interests.

Alwin Hylkema conceived and designed the experiments, performed the experiments, analyzed the data, prepared figures and/or tables, authored or reviewed drafts of the article, and approved the final draft.

Oliver J. Klokman performed the experiments, authored or reviewed drafts of the article, and approved the final draft.

The following information was supplied relating to field study approvals (i.e., approving body and any reference numbers):

Field experiments were approved by the Saba Conservation Foundation, management authority of the Saba Marine Park.

The following information was supplied regarding data availability:

The raw data is available in the Supplementary File.

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
