# Peer review of "Factors constraining natural recovery of Diadema antillarum following a mass die-off: a case study near the island of Saba, Caribbean Netherlands"

_PeerJ, doi:10.7717/peerj.20418_

## Round 0.1 · original submission · Major Revisions

· Academic Editor

Major Revisions

Dear authors,

I have now received two reviews of your manuscript, one recommending a major revision and the other advocating a minor revision. Both reviewers have raised several questions and comments, and I believe that responding to them can really improve the quality of your manuscript. Based on their feedback and my personal opinion, I invite you to revise your manuscript in line with the reviews you have received.

Please pay particular attention to shortening the abstract and restructuring the introduction by improving the background and aims of the study. Reviewer 2 raised several good points that could be addressed in the discussion to remove the ambiguities that may arise later for the reader. Specific comments on language issues also need to be clarified.

Please refer to the species in question as D. antillarum (instead of Diadema) in the revised text, as it is not the monospecific genus. In addition, consider adding a word “island” in the title, as not everyone is familiar with the location (Saba Island).
Please include with the revised version of your manuscript a detailed response to the comments received (point by point) with clear answers and references to corrections.

I look forward to receiving the revised version.
Kind regards,
Olja Vidjak

·

Basic reporting

The introduction has flow; however, define specific terms where appropriate.

Experimental design

Maybe cut down on subheadings and also references in the methods section, where appropriate.

Validity of the findings

Excellent reporting of the results. I like the fact that the authors also report no significant changes where appropriate.

Additional comments

The methods section could be more compact with fewer subheadings, however, I understand that many of the methods used were essential to the study.
The discussion section mentioned a few results that were not necessarily understood, and this was good to note, as sometimes many results are not yet well understood and require further investigation.

Reviewer 2 ·

Basic reporting

The wording and English need improvement.

A few examples:

- Introduction needs restructuring – the authors do say that Diadema is a key species, but it’s not clear exactly why turf and macroalgae are unwanted (line 67)
- Scientific language and sentence structure could be improved, eg, would recommend not to start a paragraph or even a sentence with “to which” (line 97)
- Sentence needs work (line 213): “Concurrently, the biomass per species was weighted.” Change to something like “Biomass for each species was calculated using the average fraction of echinoid material in stomach content
- Line 151 “To select Diadema predators… “ would read better as “ Diadema predators were selected from previous studies using stomach contents analyses (refs)…’’

Experimental design

The work is within the aims and scope of the journal; however, generally, the discussion needs to be broadened, and the introduction needs work and restructuring. There are also some problems with the experimental design.

Abstract needs to say "sea urchin" at least once - too many results in Abstract, without clearly indicating the main finding. The abstract is too long.

The introduction needs restructuring as the motivation for the work isn't clear, nor are the aims or background.

Some examples:

- The research question was not well defined, was not clear immediately if we were looking at recovery from the die off in the 80s or the more recent die off, and exactly why we are asked to consider both. Also, it was not immediately clear WHY the die off occurred, i.e., why did they die? Is this likely to happen again? Is it happening in other places around the world and with increasing frequency, or does it depend on location, habitat, temperature, etc? (line 53)
- Again – what caused the most recent die off (line 101)
- (just found answer in the discussion line 355- please mention earlier)
- What kind of area was affected by the die-off? Were all study sites affected?
- Somewhere in the abstract and early in the introduction, it would be good to state that this study aimed to study recovery from the most recent die-off, to gain more information about die-offs in general, and specifically in relation to the 1983/84 die-off
- There should be a paragraph about why die off would occur, and clearly that the Diadema plays a crucial role in maintaining the health of a coral reef ecosystem by grazing on algae, without this species, x, y, and z would occur.
- Why were these sites chosen? (line 111)
- How long do larvae disperse during their pelagic phase? (line 139)
- Line 211 paragraph – I’m not sure about the predator weighting, if there are fewer sea urchins, then it doesn’t make sense to weight the predation on sea urchins by fish by a stomach contents study conducted nearly 80 years ago. Possibly, this could be framed as “potential” predation
- Would be good to see results combined – i.e., show Diadema density vs shelter location,

Validity of the findings

The study is not well placed in the international context in the discussion.


Some comments:
- Line 371 – Could the lack of wrasse (or low densities of wrasse) alone be the most important factor for Diadema recovery? Why were there low densities of wrasse here?
- Line 383 paragraph – stomach contents analysis alone has its disadvantages that could be discussed. I agree that using the Randall results probably doesn’t make sense for a study conducted almost 80 years later, after multiple die-offs. I would try to approach the predator effect in a different way, possibly just recording predator densities and then inferring which ones might be important given recovery at Diadema City, and THEN comparing to older studies and whether this differed from Randall’s findings.
- What about food for Diadema – and how did this differ between the sites? Was there ample potential food/algae for them to forage at each of the sites? What else would make the artificial reef better?

---

## Round 0.2 · Minor Revisions

· Academic Editor

Minor Revisions

Dear authors, thank you for the corrections and additions to the original manuscript and for carefully considering the reviewers' suggestions. This has resulted in an improved version of the manuscript. The reviewer invited for the second round of review has, in my opinion, provided helpful comments and suggestions that could potentially increase the impact of your study. I therefore invite you to consider these in the final round of review. Please clarify the questions raised regarding Table 1 and the differences in the abundance of macropredators (Fig. 4F). As for the distribution of your findings between the main text and supplementary material, I leave this to your discretion. However, I support the reviewers' suggestion to expand on the fish pressure.

I look forward to the final version.

Kind regards,
Olja Vidjak

Reviewer 2 ·

Basic reporting

Much improved, good work.

minor wording suggestions:

Line 55 – change "which reduced population densities with 98%" to “which reduced population densities by 98%..”
Line 294 --- change "which did not differ among each other" to “ which did not differ significantly.’
Line 410 -- change "grunt, which increased the predation pressure" to “grunt, therefore increasing the predation pressure…”

Other
Results
Figure 2 – please indicate what year these pictures were taken and pref which depth

Fig 4 – need to say what the error bars represent in the figure caption

Experimental design

Is Table 1 a result of your study? Or is it the results from the Randall study? If it’s results from Randall study then I wouldn’t include it here, just say species were selected from that study. You could say, of the xx species studied in Randall (1967), we found the following species. You could show a fig of predator count and size, and point out the 3 most common species at each site or something like that.
 I can see that you do refer to this information later, so could keep the table, but would also be good to know a bit more about the fish predator species present.
 Line 317 – refers to Table S2, which contains this predator information. I would suggest including it in the paper. – you could actually combine it all into the 1 table, having the Table 1, then a column for each site showing predation pressure for each species

Line 346 – that is strange that macro-predator abundance was found to not be significant, because it does LOOK significant in fig 4F

Validity of the findings

Discussion –
Line 389 paragraph – would be good to mention the time of year this species spawns, and pelagic larval duration, as well as time of year settlement was measured with collectors to make sure it lined up. Could mention in methods.

- some of findings are linked to material in the supplementary - relating to the actual fish species found at each site. These results are interesting and could be a main driver to the conclusions. I suggest as in previous review section to include all species of fish in a table as outlined above.

I also suggest the following:

In general – it’s clear from fig 4 that fish pressure is much lower at Diadema, despite results from roving diver survey which you explain by the higher presence of grunts and lower densities of wrasse. I think higher fish predation pressure at Diadema is an important finding and should be emphasised a bit more. Possibly you could test the Randall %s but then also test another scenario where those are dialed down based on recent studies – or most simply (and preferred) just show densities of what you think are the 3 or 4 main fish species predators that still prey on D. antillarium, eg show total fish density combined of wrasse, queen triggerfish, Spanish hogfish at all sites and don’t scale by stomach content %. This would be an interesting finding and you are already discussing this finding, so it would be good to support it with a figure (could add to fig 4 – potentially could remove roving diver survey as this was not at all significant and likely biased by grunt)

Additional comments

The manuscript has been greatly improved and flows a lot better now. I have some minor revisions, but I believe all the pieces are there. I'd just like to see some more detail around fish predator abundances and your critical thinking around which of those could be the more important predators for the urchin, then comparing their densities at the 4 sites. This could strengthen your finding and the discussion points you have already carefully laid out and supported.

---

## Round 0.3 · accepted · Accept

· Academic Editor

Accept

Dear authors,

Thank you for the final review and for considering the reviewer's suggestions. In terms of content, I consider this the final version, ready for publication. Upon final reading, I noticed some minor formatting issues. Please ensure that all species names are italicised throughout the manuscript (e.g. Agaricia, Orbicolla; Cinetorhynchus manningi, antillarum) and that the typo in the Latin name of bluehead is corrected (bifasciatum, not fbifasciatum).

Congratulations on your work and thank you for choosing PeerJ as your publishing platform.

Kind regards,
Olja Vidjak